# UNIPINNS: A UNIFIED PINNS FRAMEWORK FOR MULTI-TASK LEARNING OF DIVERSE NAVIER-STOKES EQUATIONS

## ABSTRACT

In recent years, Physics-Informed Neural Networks (PINN) have been used for flow simulations of various incompressible Navier-Stokes equations, but single-task PINNs have inherent limitations: low data efficiency under sparse supervision, weak generalization ability across flow patterns, and high computational costs due to repeated training. Furthermore, inter-task negative transfer constrains their performance in complex flow simulations. To tackle these challenges, this study proposes a multi-task PINN framework that combines cross-task attention with dynamic weight allocation (DWA) strategies. The aim is to verify the efficiency of this framework in various typical flows, focusing on alleviating negative transfer, enhancing training stability across different viscosities, and exploring the range of advantages. Extensive experiments demonstrate the effectiveness of our approach. Especially, with the cross-task attention module, inter-task negative transfer is significantly mitigated, enabling flow tasks with advantages to maintain stable training curves; the introduction of dynamic weight allocation further reduces loss oscillations during training, notably enhancing the convergence speed of certain flow tasks.

## 1 INTRODUCTION

Physics-Informed Neural Networks (PINNs) have demonstrated revolutionary potential in solving partial differential equations since their inception(Raissi et al., 2019), particularly achieving remarkable success in fluid mechanics. The core innovation lies in directly embedding physical laws such as momentum conservation and mass conservation into neural network loss functions, utilizing automatic differentiation to compute high-order derivatives, and ensuring networks satisfy physical constraints during training. This approach enables learning complex fluid dynamics behaviors from sparse observational data and handling complex geometries and boundary conditions that are challenging for traditional numerical methods. In single-flow scenarios, PINNs have been successfully applied to classic fluid problems including lid-driven cavity flow(Bai et al., 2020), pipe flow(Urbanowicz et al., 2023), and couette flow(Mehta et al., 2019), accurately capturing vortex structures, predicting laminar-to-turbulent transitions, and solving velocity distributions and pressure fields at different Reynolds numbers, demonstrating their tremendous potential in fluid mechanics computation.

However, real-world fluid problems often involve multiple flow types, each with unique physical characteristics and boundary conditions, presenting new challenges for PINN applications. Multi-flow learning faces differences in physical parameters, boundary conditions, flow field characteristics, and geometric configurations. Different flow types have varying physical parameters (such as viscosity, density, and characteristic velocity) that affect the relative importance of convective and diffusive terms in the Navier-Stokes equations, leading to fundamental differences in flow field characteristics(Málek & Rajagopal, 2005).

Furthermore, existing methods often treat different flow types as independent problems, training separate networks for each flow type. This approach not only incurs high computational costs but also ignores physical similarities and knowledge transfer potential between flow types. Different flow types share similar characteristics in boundary layer velocity gradients and pressure distributions

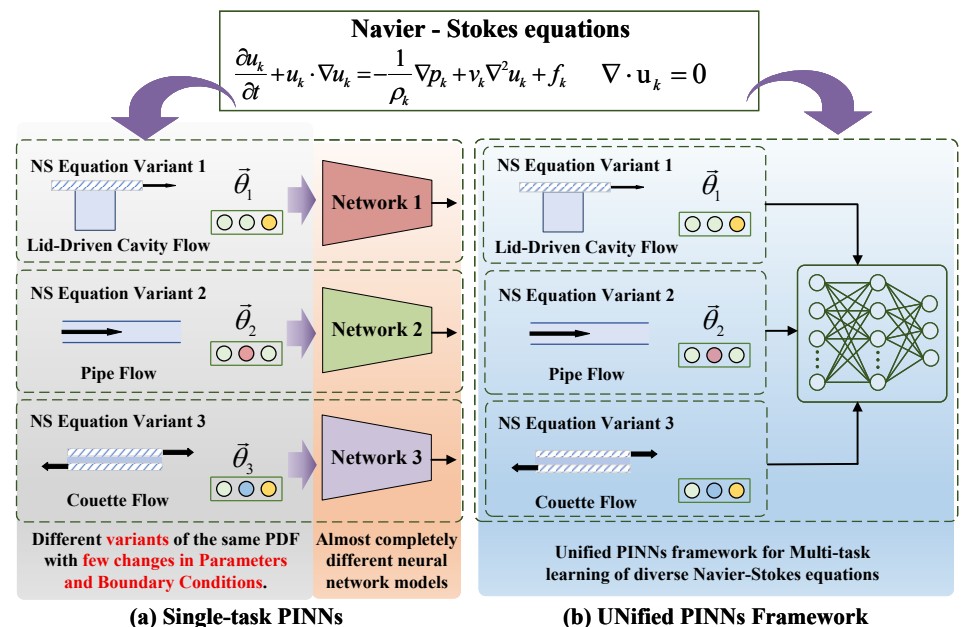

Figure 1: **Single-task vs. Unified PINNs Framework:** (a) Single-task PINNs train independent networks for each fluid problem, leading to parameter redundancy; (b) Unified PINNs framework uses shared networks to simultaneously learn multiple fluid problems.

at vortex centers(Tanarro et al., 2020), yet these features cannot be effectively utilized by independently trained networks. As shown in Fig. 2, in reality, all flow types follow the Navier-Stokes equations, differing mainly in boundary and initial conditions, and all involve fundamental physical principles such as mass conservation, momentum conservation, and energy conservation, potentially exhibiting similar flow instabilities and transition phenomena(Caetano, 2025). These commonalities provide an important foundation for cross-flow knowledge sharing, but existing methods fail to effectively utilize them, lacking unified multi-flow learning architectures.

To address these challenges, this paper proposes a unified multi-flow Physics-Informed Neural Network framework (UniPINNs) that embeds physical constraints from different flow types into a unified loss function through shared-specialized network architectures and cross-flow attention mechanisms, while employing adaptive weight balancing strategies to handle multi-flow parameter differences, achieving cross-flow knowledge sharing and efficient learning. The main contributions of this paper are as follows:

**(1) Unified multi-flow learning framework:** This paper constructs a unified multi-flow learning framework using shared-specialized network architectures that can simultaneously handle multiple flow types, significantly improving training efficiency.

**(2) Cross-flow attention mechanism design:** This paper designs a cross-flow attention mechanism that enables knowledge interaction and transfer between flow types through self-attention and cross-flow attention modules, identifying similar physical patterns across different flow types.

**(3) Adaptive weight balancing strategy:** This paper proposes an adaptive weight balancing strategy based on dynamic weight adjustment that automatically adjusts loss weights according to the training state of each flow type.

## 2 RELATED WORK

**Physics-Informed Neural Networks.** Since their introduction, PINNs have achieved significant development in both theoretical foundations and practical applications. The core idea of PINNs is to embed physical laws into neural network loss functions, using automatic differentiation to compute high-order derivatives, thereby forcing networks to satisfy physical constraints during train-

ing(Raissi et al., 2019). This method enables learning complex physical phenomena from sparse observational data and handles complex geometries and boundary conditions that are difficult for traditional numerical methods. In fluid mechanics, [23] successfully applied PINNs to incompressible Navier-Stokes equations. In recent years, some studies have begun exploring PINN applications to more complex fluid problems(Zhao et al., 2024), but most methods still focus primarily on single flow types(Wong et al., 2024), lacking cross-flow knowledge sharing capabilities.

**Multi-Task Learning.** Multi-Task Learning (MTL) is an important research direction in machine learning(Chen et al., 2024), aiming to improve overall performance by simultaneously learning multiple related tasks(Zhang & Yang, 2021). The core idea of MTL is to utilize similarities and correlations between tasks to improve performance through shared representation learning. Caruana proposed the basic MTL framework in 1997(Caruana, 1997), and provided a comprehensive survey of MTL in 2017(Ruder, 2017). In recent years, some studies have begun exploring more complex MTL methods, such as uncertainty-weighted multi-task learning approaches that can automatically adjust weights for different tasks(Chen et al., 2025). Duan proposed adaptive multi-task learning frameworks(Duan & Wang, 2023), providing new theoretical foundations for multi-flow PINNs. However, traditional MTL methods primarily focus on supervised learning tasks, lacking consideration of physical constraints.

**Attention Mechanisms.** Attention mechanisms are an important technology in deep learning that can automatically learn the importance of different positions in input sequences. The Transformer model proposed in 2017 elevated attention mechanisms to new heights(Vaswani et al., 2017), achieving breakthrough progress in natural language processing and other fields. In computer vision, attention mechanisms are widely applied to image classification, object detection, and other tasks(Guo et al., 2022). Non-local neural networks can capture long-distance dependencies through self-attention mechanisms(Wang et al., 2018). Vision Transformer successfully applied attention mechanisms to image classification tasks(Maurício et al., 2023). However, existing attention mechanisms in PINNs applications primarily focus on single modalities or single tasks, lacking cross-task knowledge sharing capabilities.

**Adaptive Weight Balancing.** In multi-task learning, balancing loss weights for different tasks is an important problem(Kirchdorfer et al., 2024). Traditional weight balancing methods mainly include fixed weights and manual weight tuning, which often require extensive experimentation and tuning, making them difficult to adapt to different task variations. In recent years, some studies have begun exploring adaptive weight balancing methods. Uncertainty-weighted multi-task learning methods proposed in 2019 can automatically adjust weights for different tasks(Gong et al., 2019). Gradient-balanced multi-task learning methods proposed in 2024 can avoid overfitting in certain tasks(Lee & Kim, 2023). Adaptive weight balancing strategies provide new solutions for multi-flow PINNs(Kumar & Yadav, 2025). However, existing adaptive weight balancing methods primarily focus on supervised learning tasks, lacking consideration of physical constraints.

Through analysis of related work, it can be seen that physics-informed neural networks have achieved significant progress in single-flow problems, but still face numerous challenges in multi-flow learning. Traditional multi-flow computational methods are primarily based on numerical methods with high computational complexity and lack learning capabilities. Existing multi-task learning methods can handle multiple related tasks but lack consideration of physical constraints. Attention mechanisms can capture complex relationships but primarily focus on single modalities or single tasks. Adaptive weight balancing methods can automatically adjust weights but lack consideration of physical constraints. These limitations provide important opportunities for this research. The UniPINNs framework proposed in this paper combines PINNs with multi-task learning, attention mechanisms, and adaptive weight balancing technologies to achieve cross-flow knowledge sharing and efficient learning.

## 3 METHODOLOGY

### 3.1 PROBLEM DEFINITION

We consider the problem of learning multiple incompressible Navier-Stokes flow patterns simultaneously using Physics-Informed Neural Networks. Given a set of flow types $\mathcal{F} = \{f_1, f_2, \ldots, f_k\}$, where each flow type $f_k$ is characterized by its specific physical parameters (e.g., viscosity coeffi-

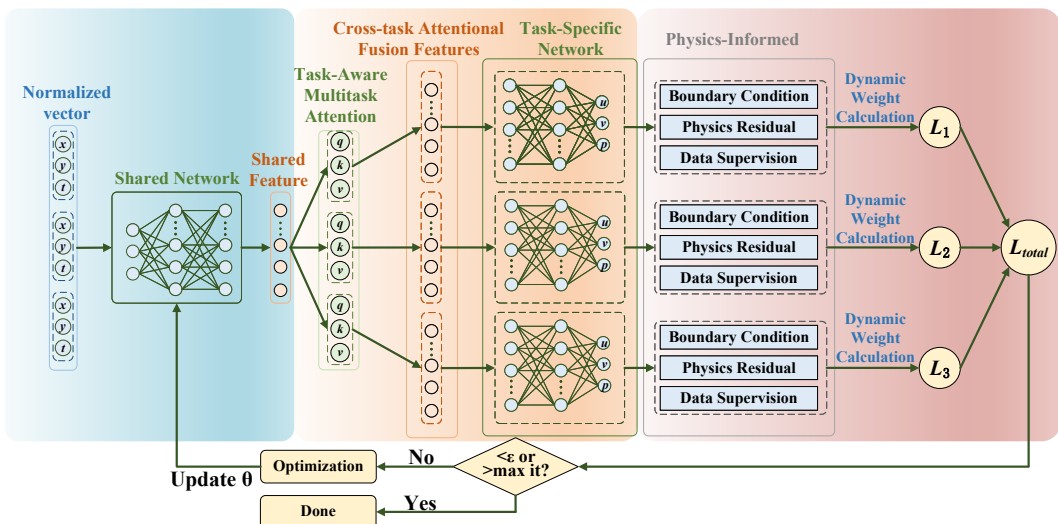

Figure 2: **Unified Multi-Flow Learning Framework.** The framework consists of three components: (1) shared feature extraction and attention fusion; (2) parallel task-specific processing; (3) physics-informed constraints and optimization.

cient $\nu_k$, density $\rho_k$) and boundary conditions, our goal is to train a unified neural network that can accurately predict the velocity field $\mathbf{u}_k(x,t)$ and pressure field $p_k(x,t)$ for any flow type $f_k$.

For each flow type $f_k$, the governing equations are the incompressible Navier-Stokes equations(Gu et al., 2024):

$$\frac{\partial \mathbf{u}_k}{\partial t} + \mathbf{u}_k \cdot \nabla \mathbf{u}_k = -\frac{1}{\rho_k}\nabla p_k + \nu_k \nabla^2 \mathbf{u}_k + \mathbf{f}_k, \tag{1}$$

$$\nabla \cdot \mathbf{u}_k = 0, \tag{2}$$

where $\mathbf{u}_k(x,t) \in \mathbb{R}^2$ is the velocity field of flow type $f_k$ at spatial position $x = (x,y)$ and time $t$, $p_k(x,t) \in \mathbb{R}$ is the pressure field, $\mathbf{f}_k(x,t) \in \mathbb{R}^2$ is the body force, $\rho_k \in \mathbb{R}^+$ is the density, $\nu_k \in \mathbb{R}^+$ is the kinematic viscosity coefficient, and the spatial domain $\Omega_k \subset \mathbb{R}^2$ and time domain $[0, T_k] \subset \mathbb{R}^+$ are flow-type specific.

### 3.2 Unified Multi-Flow Learning Framework

The proposed UniPINNs framework adopts a hierarchical architecture design, consisting of three core components: a shared backbone network, task-specific dedicated layers, and a cross-flow attention mechanism, as shown in Fig. 2. This design ensures both knowledge sharing between flow types and maintains the specificity of each flow type.

**Shared Backbone Network.** The shared backbone network $\mathcal{N}_s$ is the core of the entire framework, responsible for learning common physical features across all flow types. This network adopts a multi-layer perceptron structure, receiving normalized input coordinates $(x, y, t)$ and flow type indicator $k$ as input. Through deep feature extraction, the network outputs intermediate features $\mathbf{h} = \mathcal{N}_s(x, y, t, k)$, which can capture universal fluid dynamics patterns such as mass conservation, momentum conservation, and other fundamental physical principles. The design of the shared network enables different flow types to share these common features, improving learning efficiency.

**Task-Specific Dedicated Layers.** Each flow type $f_k$ has its own independent dedicated layer $\mathcal{N}_k$, responsible for converting shared features into flow-specific predictions. The dedicated layers not only process shared features $\mathbf{h}$ but also explicitly incorporate flow-specific physical parameters $\nu_k$ and $\rho_k$. $\nu_k$ is the kinematic viscosity coefficient, characterizing the viscous properties of the fluid and determining the intensity of momentum diffusion; $\rho_k$ is the fluid density, characterizing the inertial properties of the fluid and determining the mass distribution of the fluid. This design allows the network to adjust its behavior according to various physical parameters, ensuring that predictions

conform to the physical characteristics of each flow type. The output of the dedicated layers is:

$$\mathbf{u}_k, p_k = \mathcal{N}_k(\mathbf{h}, \nu_k, \rho_k), \tag{3}$$

where $\mathbf{u}_k \in \mathbb{R}^2$ is the predicted velocity field and $p_k \in \mathbb{R}$ is the predicted pressure field.

**Cross-Flow Attention Mechanism.** The attention mechanism $\mathcal{A}$ is an innovative component of the framework, achieving knowledge transfer by computing similarities between different flow type features. This mechanism can identify similar physical patterns in key regions across different flow types, thereby achieving true cross-flow knowledge sharing:

$$\mathbf{h}'_k = \mathcal{A}(\mathbf{h}_k, \{\mathbf{h}_j\}_{j=1}^K), \tag{4}$$

where $\mathbf{h}'_k \in \mathbb{R}^d$ is the enhanced feature for flow type $k$ after incorporating information from other flow types, and $d$ is the feature dimension.

### 3.3 CROSS-FLOW ATTENTION MECHANISM

The cross-flow attention mechanism achieves knowledge interaction and transfer between flow types through carefully designed attention mechanisms. This mechanism consists of self-attention modules and cross-attention modules, effectively capturing common features and differences between flow types.

**Self-Attention Module.** The self-attention module first performs internal refinement of features for each flow type. For flow type $f_k$, the query matrix $\mathbf{Q}_k$, key matrix $\mathbf{K}_k$, and value matrix $\mathbf{V}_k$ are computed as follows:

$$\mathbf{Q}_k = \mathbf{h}_k \mathbf{W}_Q, \tag{5}$$
$$\mathbf{K}_k = \mathbf{h}_k \mathbf{W}_K, \tag{6}$$
$$\mathbf{V}_k = \mathbf{h}_k \mathbf{W}_V, \tag{7}$$

where $\mathbf{W}_Q \in \mathbb{R}^{d \times d_q}$, $\mathbf{W}_K \in \mathbb{R}^{d \times d_k}$, $\mathbf{W}_V \in \mathbb{R}^{d \times d_v}$ are learnable weight matrices, and $d_q$, $d_k$, $d_v$ are the dimensions of query, key, and value, respectively. The attention weights are computed through the softmax function:

$$\text{Attention}_k = \text{softmax}\left(\frac{\mathbf{Q}_k \mathbf{K}_k^T}{\sqrt{d_k}}\right), \tag{8}$$

where $d_k$ is the key dimension, used to scale attention scores and prevent gradient vanishing. The final self-attention output is:

$$\mathbf{h}_k^{self} = \text{Attention}_k \mathbf{V}_k. \tag{9}$$

The self-attention module can capture long-range dependencies within flow types and identify important physical features.

**Cross-Attention Module.** The cross-attention module achieves knowledge interaction between flow types. For flow type $k$, the attention between its query matrix and other flow types' key-value matrices is computed as follows:

$$\mathbf{Q}_k^{cross} = \mathbf{h}_k^{self} \mathbf{W}_Q^{cross}, \quad \mathbf{K}_j^{cross} = \mathbf{h}_j^{self} \mathbf{W}_K^{cross}, \quad \mathbf{V}_j^{cross} = \mathbf{h}_j^{self} \mathbf{W}_V^{cross}, \tag{10}$$

where $j = 1, 2, \ldots, K$ represents the index of other flow types, and $\mathbf{W}_Q^{cross} \in \mathbb{R}^{d \times d_q^{cross}}$, $\mathbf{W}_K^{cross} \in \mathbb{R}^{d \times d_k^{cross}}$, $\mathbf{W}_V^{cross} \in \mathbb{R}^{d \times d_v^{cross}}$ are learnable weight matrices for cross-attention. The cross-attention weights are computed as:

$$\text{CrossAttention}_{k,j} = \text{softmax}\left(\frac{\mathbf{Q}_k^{cross}(\mathbf{K}_j^{cross})^T}{\sqrt{d_{cross}}}\right), \tag{11}$$

where $d_{cross}$ is the key dimension for cross-attention. The final cross-attention output is the weighted sum of all flow type features:

$$\mathbf{h}_k^{cross} = \sum_{j=1}^K \text{CrossAttention}_{k,j} \mathbf{V}_j^{cross}. \tag{12}$$

**Feature Fusion.** To balance self-features and cross-flow information, a weighted fusion strategy is adopted:

$$\mathbf{h}_k^{enhanced} = \alpha \mathbf{h}_k^{self} + (1-\alpha)\mathbf{h}_k^{cross}, \tag{13}$$

where $\alpha \in [0,1]$ is a learnable parameter controlling the relative importance of the two types of information. This design enables the network to dynamically adjust its dependence on self-features and cross-flow information during training.

## 3.4 ADAPTIVE WEIGHT BALANCING STRATEGY

A key challenge in multi-flow learning is how to balance loss weights across different flow types. Traditional fixed weight methods cannot adapt to the convergence characteristics of different flow types. Therefore, we propose an adaptive weight balancing strategy based on dynamic weight adjustment.

**Loss Component Analysis.** For each flow type $f_k$, the total loss consists of three main components:

$$\mathcal{L}_k = \mathcal{L}_{eq,k} + \mathcal{L}_{bc,k} + \mathcal{L}_{data,k}, \tag{14}$$

where $\mathcal{L}_{eq,k}$ is the equation residual loss, $\mathcal{L}_{bc,k}$ is the boundary condition loss, and $\mathcal{L}_{data,k}$ is the data supervision loss.

**Equation Residual Loss.** The equation residual loss is defined as:

$$\mathcal{L}_{eq,k} = \frac{1}{|\Omega_k|} \int_{\Omega_k} \left| \frac{\partial \mathbf{u}_k}{\partial t} + \mathbf{u}_k \cdot \nabla \mathbf{u}_k + \frac{1}{\rho_k} \nabla p_k - \nu_k \nabla^2 \mathbf{u}_k - \mathbf{f}_k \right|^2 dx, \tag{15}$$

where $|\Omega_k|$ is the area of domain $\Omega_k$.

**Boundary Condition Loss.** The boundary condition loss is defined as:

$$\mathcal{L}_{bc,k} = \frac{1}{|\partial \Omega_k|} \int_{\partial \Omega_k} \left[ |\mathbf{u}_k - \mathbf{u}_{k,bc}|^2 + |p_k - p_{k,bc}|^2 \right] ds, \tag{16}$$

where $\partial \Omega_k$ is the boundary of domain $\Omega_k$, $|\partial \Omega_k|$ is the length of the boundary, and $\mathbf{u}_{k,bc}$ and $p_{k,bc}$ are the given values on the boundary.

**Data Supervision Loss.** The data supervision loss is defined as:

$$\mathcal{L}_{data,k} = \frac{1}{|D_k|} \sum_{(x,t) \in D_k} \left[ |\mathbf{u}_k(x,t) - \mathbf{u}_{k,obs}(x,t)|^2 + |p_k(x,t) - p_{k,obs}(x,t)|^2 \right], \tag{17}$$

where $D_k$ is the observation dataset for flow type $f_k$, $|D_k|$ is the number of observation points, and $\mathbf{u}_{k,obs}$ and $p_{k,obs}$ are the observed values.

**Dynamic Weight Calculation.** The core idea of adaptive weights is to dynamically adjust weights based on the training progress of each flow type. The relative improvement rate is defined as:

$$r_k^{(t)} = \frac{\mathcal{L}_k^{(t-1)} - \mathcal{L}_k^{(t)}}{\mathcal{L}_k^{(t-1)}}, \tag{18}$$

which reflects the training progress of flow type $k$ at time step $t$. Based on the relative improvement rate, the adaptive weights are computed as:

$$w_k^{(t)} = \frac{\exp(\beta \cdot r_k^{(t)})}{\sum_{j=1}^{K} \exp(\beta \cdot r_j^{(t)})}, \tag{19}$$

where $\beta > 0$ is the temperature parameter controlling the sharpness of the weight distribution, and $j = 1, 2, \ldots, K$ represents the index of all flow types. When $\beta$ is large, weights are more concentrated on flow types with faster training progress; when $\beta$ is small, the weight distribution is more uniform.

**Weight Smoothing Mechanism.** To avoid violent weight fluctuations, an exponential moving average mechanism is introduced:

$$\tilde{w}_k^{(t)} = \gamma \tilde{w}_k^{(t-1)} + (1-\gamma) w_k^{(t)}, \tag{20}$$

where $\gamma \in [0, 1]$ is the smoothing parameter, typically set to 0.9.

**Multi-Flow Total Loss.** The final multi-flow total loss is computed as:

$$\mathcal{L}_{total} = \sum_{k=1}^{K} \tilde{w}_k^{(t)} \mathcal{L}_k. \tag{21}$$

# 4  EXPERIMENTS AND RESULTS

To comprehensively validate the effectiveness of the proposed UniPINNs framework, we designed multi-level experiments based on the PDEBench dataset to evaluate cross-flow learning capabilities, physical constraint satisfaction, and computational efficiency.

## 4.1  DATASET DESCRIPTION

We employ three classic incompressible Navier-Stokes flow types from the PDEBench dataset as test benchmarks, each characterized by distinct physical properties and boundary conditions:

**Lid-Driven Cavity Flow**: Features fixed geometric boundaries and top-driven conditions, involving Dirichlet boundary conditions. This flow type is characterized by vortex structures, including complex interactions among primary, secondary, and tertiary vortices, as well as boundary layer development and evolution processes.

**Pipe Flow**: Involves inlet-outlet boundary conditions requiring Neumann boundary condition handling. This flow type is characterized by boundary layer development and velocity profiles, including entrance effects, fully developed flow, and velocity profile evolution. Pipe flow exhibits unique geometric configurations and inlet velocity distribution characteristics.

**Couette Flow**: Features relatively moving wall boundaries involving mixed boundary conditions. This flow type is characterized by linear velocity distributions and pressure gradients, including wall effects and shear stress distributions. Couette flow demonstrates flow characteristics under different wall velocity ratios.

The three flow types exhibit significant differences in physical parameters, boundary condition types, flow field characteristic patterns, and geometric configurations, providing an ideal test platform for validating cross-flow knowledge sharing. All flow types follow the same Navier-Stokes equations but differ in boundary and initial conditions, providing a physical foundation for cross-flow learning.

## 4.2  EXPERIMENTAL CONFIGURATION AND EVALUATION METRICS

The experiments employ a shared backbone network and dedicated layers with 3 layers × 64 neurons, using Tanh activation functions. The attention mechanism comprises 8 attention heads, each with a dimension of 64. Training uses the Adam optimizer with a joint training learning rate of $1 \times 10^{-3}$, StepLR decay strategy (step size 1000, decay factor 0.85), and batch size of 1024. All experiments are conducted on an NVIDIA RTX 4090 GPU with 24GB of memory, using mean squared error as the primary evaluation metric to measure the satisfaction of physical constraints and prediction accuracy.

## 4.3  EXPERIMENTAL RESULTS AND VISUALIZATION ANALYSIS

Fig. 3 displays streamline patterns for three typical fluid flows, validating the UniPINNs framework's accurate prediction capability for different flow types. Fig. 4 shows the loss curves during multi-flow PINN training, where all three flow types exhibit rapid loss reduction in the early training phase followed by stabilization. Comparative analysis reveals that pipe flow shows higher loss values across all loss terms, primarily due to its complex Neumann boundary conditions and inlet effects; Couette flow typically achieves the lowest loss levels, benefiting from its relatively simple linear velocity distribution and mixed boundary condition characteristics; lid-driven cavity flow performs excellently in equation loss and boundary condition loss but has slightly higher total loss than Couette flow, mainly attributed to its complex vortex structures and the difficulty of handling

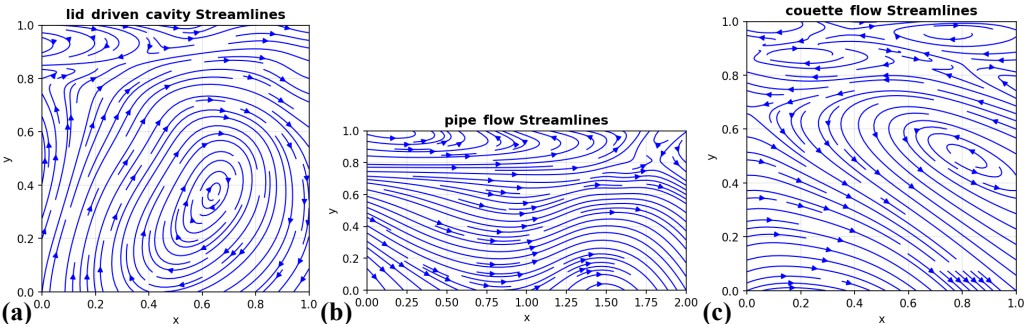

Figure 3: **Streamline Patterns of Three Typical Fluid Flows.** This figure presents streamline patterns of three typical fluid flows: (a) lid-driven cavity flow, (b) pipe flow, and (c) Couette flow. All streamlines are shown in blue with arrows indicating flow direction.

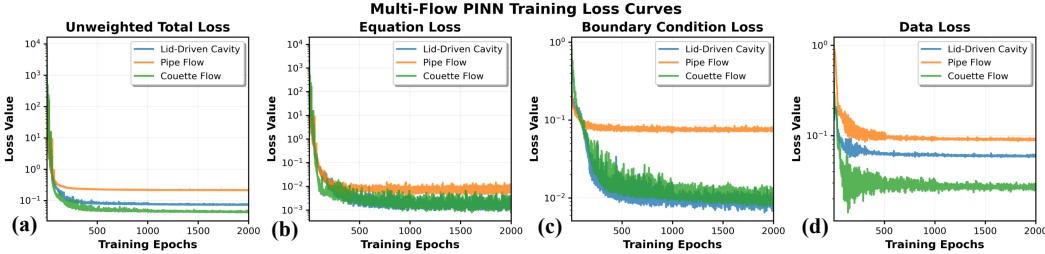

Figure 4: **Multi-Flow PINN Training Loss Curves.** This figure presents the training loss curves of Multi-Flow PINN: (a) unweighted total loss, (b) equation loss, (c) boundary condition loss, and (d) data loss.

Dirichlet boundary conditions. These results demonstrate that our unified framework can effectively handle physical characteristic differences across flow types, achieving balanced optimization of multi-task learning through dynamic weight adjustment.

## 4.4 COMPARATIVE EXPERIMENTS

As shown in Table 1, UniPINNs achieves the best performance across all three flow types, validating the effectiveness of the unified multi-flow learning framework. In lid-driven cavity flow, UniPINNs achieves an error of 1.27e-02, improving by 32% compared to single-task PINN (1.87e-02), 49% compared to Gaussian process regression (2.51e-02), and 81% compared to linear regression (6.61e-02). In pipe flow, UniPINNs achieves an error of 1.25e-01, improving by 8% compared to single-task PINN (1.36e-01), 57% compared to ALPINN (2.91e-01), and 90% compared to MMPDE-Net (1.26e+00). In Couette flow, UniPINNs achieves an error of 1.27e-02, improving by 24% compared to single-task PINN (1.68e-02), 84% compared to ALPINN (7.73e-02), and 98% compared to MMPDE-Net (5.78e-01).

**Compared to the latest PINN methods**: UniPINNs achieves multi-flow unified learning while maintaining high accuracy. The ALPINN method shows lower performance than our method across all three flow types, particularly in pipe flow, where the error reaches 2.91e-01, 133% higher than UniPINNs. Although MMPDE-Net has evaluation results across all flow types, its performance is generally low, with pipe flow error reaching 1.26, 908% higher than UniPINNs, indicating its limitations in large-scale multi-flow learning. Compared to single-task PINN, UniPINNs achieves performance improvement through multi-task learning mechanisms, validating the effectiveness of cross-flow knowledge sharing, where multi-task learning not only maintains performance but also enhances overall performance through feature sharing and regularization effects.

**Task-specific method analysis**: LAAF, DNN, and KIH-PINN task-specific methods are only evaluated on pipe flow, with performance of 2.26e-01, 8.00e-02, and 9.41e-01, respectively. Although DNN performs relatively well on pipe flow, its error is still 540% higher than UniPINNs. These

Table 1: Comparative Study Results: Performance Comparison Across Different Methods

| Method | Lid-Driven Cavity | Pipe Flow | Couette Flow |
|---|---|---|---|
| Linear Regression | 6.61e-02 | - | - |
| Gaussian Process Regression | 2.51e-02 | - | - |
| LAAF | - | 2.26e-01 | - |
| DNN | - | 8.00e-02 | - |
| KIH-PINN | - | 9.41e-01 | - |
| ALPINN | 1.25e-01 | 2.91e-01 | 7.73e-02 |
| MMPDE-Net | 7.68e-01 | 1.26e+00 | 5.78e-01 |
| Single-Task PINN | 1.87e-02 | 1.36e-01 | 1.68e-02 |
| **UniPINNs (Proposed)** | **1.27e-02** | **1.25e-01** | **1.27e-02** |

results further demonstrate the advantages of the unified framework over task-specific methods, avoiding the complexity of designing and training separate models for each flow type.

## 4.5 ABLATION STUDIES

To validate the effectiveness of each component in the UniPINNs framework, we conduct comprehensive ablation experiments by systematically removing individual components. Table 2 presents performance comparisons across different component configurations. The results show that each component contributes significantly to overall performance. Specifically, the complete model with all components achieves the best performance across all three flow types. Removing the dynamic weight adjustment (DWA) component leads to performance degradation, particularly evident in pipe flow, where the error increases from 1.25e-01 to 2.17e-01, a 74% increase, indicating the importance of adaptive weight balancing in multi-task learning scenarios. The attention mechanism has a significant impact on performance, with substantial performance drops when removed: lid-driven cavity flow error increases from 1.27e-02 to 8.59e-02, a 577% increase, while pipe flow error increases from 1.25e-01 to 2.62e-01, a 110% increase. This demonstrates that the cross-flow attention mechanism effectively captures and utilizes common features across different flow types. The ablation experiments confirm that all three components work synergistically to achieve optimal performance, with progressive performance degradation as components are removed, demonstrating the necessity of each component in our proposed framework.

Table 2: Ablation Study Results: Component Analysis and Performance Comparison

| Attention | Input Norm | DWA | Lid-Driven | Pipe Flow | Couette Flow |
|---|---|---|---|---|---|
| ✓ | ✓ | ✓ | 1.27e-02 | 1.25e-01 | 1.27e-02 |
| ✓ | ✓ | ✗ | 1.83e-02 | 2.17e-01 | 1.18e-01 |
| ✓ | ✗ | ✓ | 2.57e-02 | 1.65e-01 | 2.40e-02 |
| ✗ | ✓ | ✓ | 8.59e-02 | 2.62e-01 | 1.62e-02 |
| ✗ | ✗ | ✓ | 1.22e-01 | 3.51e-01 | 6.39e-02 |
| ✗ | ✗ | ✗ | 1.42e-01 | 3.78e-01 | 9.21e-02 |

## 5 CONCLUSION

This paper presents UniPINNs, a unified Physics-Informed Neural Network framework for multi-task learning of diverse Navier-Stokes equations. Our approach addresses the fundamental challenges in multi-flow learning by introducing cross-flow attention mechanisms, adaptive weight balancing strategies, and input normalization techniques. Our experiments on three classic Navier-Stokes flow types demonstrate the effectiveness of the proposed framework. UniPINNs achieves superior performance across all flow types, with significant improvements over existing methods. The ablation studies confirm the necessity of each component in our framework. The results demonstrate that our unified framework effectively handles physical characteristic differences across flow types, achieving balanced optimization of multi-task learning through dynamic weight adjustment. In the future, we will focus on extending the framework to more complex flow scenarios and exploring applications to other types of partial differential equations.

## ETHICS STATEMENT

This work adheres to the ICLR Code of Ethics. In this study, no human subjects or animal experimentation was involved. All datasets used, including PDEBench dataset, were sourced in compliance with relevant usage guidelines, ensuring no violation of privacy. We have taken care to avoid any biases or discriminatory outcomes in our research process. No personally identifiable information was used, and no experiments were conducted that could raise privacy or security concerns. We are committed to maintaining transparency and integrity throughout the research process.

## REPRODUCIBILITY STATEMENT

We have made every effort to ensure that the results presented in this paper are reproducible. All code and datasets have been made publicly available in an anonymous repository to facilitate replication and verification. The experimental setup, including training steps, model configurations, and hardware details, is described in detail in the paper. We have also provided a full description of our UniPINNs framework, including the cross-flow attention mechanism, adaptive weight balancing strategy, and input normalization methods, to assist others in reproducing our experiments.

Additionally, the public datasets used in the paper, such as the PDEBench dataset, are publicly available, ensuring consistent and reproducible evaluation results.

We believe these measures will enable other researchers to reproduce our work and further advance the field.

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

# A  APPENDIX

## A.1  LLM USAGE

Large Language Models (LLMs) were used to aid in the writing and polishing of the manuscript. Specifically, we used an LLM to assist in refining the language, improving readability, and ensuring clarity in various sections of the paper. The model helped with tasks such as sentence rephrasing, grammar checking, and enhancing the overall flow of the text.

It is important to note that the LLM was not involved in the ideation, research methodology, or experimental design. All research concepts, ideas, and analyses were developed and conducted by the authors. The contributions of the LLM were solely focused on improving the linguistic quality of the paper, with no involvement in the scientific content or data analysis.

The authors take full responsibility for the content of the manuscript, including any text generated or polished by the LLM. We have ensured that the LLM-generated text adheres to ethical guidelines and does not contribute to plagiarism or scientific misconduct.

## A.2  INPUT NORMALIZATION AND COORDINATE TRANSFORMATION

Different flow types have different geometric configurations and physical parameters, leading to significant differences in the scale and distribution of input data. To ensure that the network can effectively handle these differences, specialized input normalization and coordinate transformation strategies are designed.

**Coordinate Normalization Strategy.**  For each flow type $f_k$, its spatial domain is $\Omega_k = [x_{min,k}, x_{max,k}] \times [y_{min,k}, y_{max,k}]$. Coordinates are normalized to the standard interval $[0,1]^2$:

$$\tilde{x} = \frac{x - x_{min,k}}{x_{max,k} - x_{min,k}}, \tag{22}$$

$$\tilde{y} = \frac{y - y_{min,k}}{y_{max,k} - y_{min,k}}, \tag{23}$$

where $x_{min,k}$, $x_{max,k}$, $y_{min,k}$, $y_{max,k}$ are the minimum and maximum coordinate values of flow type $f_k$ in the $x$ and $y$ directions, respectively. This normalization strategy eliminates geometric scale differences between different flow types, enabling the network to focus on learning physical patterns rather than geometric scales.

**Time Normalization Method.** Time normalization is performed based on the characteristic time scale of each flow type:

$$\tilde{t} = \frac{t}{T_{char,k}}, \tag{24}$$

where $T_{char,k} = L_k/U_k$ is the characteristic time scale of flow type $f_k$, $L_k$ is the characteristic length scale, and $U_k$ is the characteristic velocity scale. This normalization ensures that the time evolution of different flow types occurs on the same scale.

**Physical Parameter Encoding.** To fully utilize flow-specific physical parameters, they are encoded as additional input features:

$$\mathbf{p}_k = [\nu_k, \rho_k, \text{Re}_k, \text{other flow-specific parameters}], \tag{25}$$

where $\text{Re}_k = U_k L_k/\nu_k$ is the Reynolds number of flow type $f_k$. These physical parameters are provided as additional input to the network, enabling the network to adjust its behavior according to different physical conditions.

**Feature Enhancement.** To further improve the network's expressive capability, feature enhancement is performed on the normalized coordinates:

$$\mathbf{x}_{enhanced} = [\tilde{x}, \tilde{y}, \tilde{t}, \sin(2\pi\tilde{x}), \cos(2\pi\tilde{x}), \sin(2\pi\tilde{y}), \cos(2\pi\tilde{y}), \mathbf{p}_k]. \tag{26}$$

This enhancement strategy introduces periodic features, helping the network capture periodic structures in the flow field.

## A.3 TRAINING STRATEGY

The training strategy adopts a phased approach, ensuring that the network can gradually learn the mapping from universal features to flow-specific features.

**Pre-training Phase.** In the pre-training phase, the shared backbone network is trained using a subset of data from all flow types. The goal of this phase is to enable the network to learn common physical features between flow types, such as mass conservation, momentum conservation, and other fundamental physical principles. Pre-training uses a smaller learning rate and fewer training epochs to avoid overfitting.

**Joint Training Phase.** The joint training phase is the core of the entire framework, using adaptive weight balancing strategy to train the complete UniPINNs framework end-to-end. In this phase, the cross-flow attention mechanism is gradually activated, and the network begins to learn knowledge interaction between flow types. During training, the convergence of each flow type is monitored, and weight allocation is dynamically adjusted.

**Fine-tuning Phase.** The fine-tuning phase optimizes performance for specific flow types. In this phase, the shared network parameters remain unchanged, and only the dedicated layers of each flow type are fine-tuned. This strategy maintains knowledge sharing between flow types while allowing each flow type to optimize according to its specific requirements.

**Training Monitoring.** Throughout the training process, the loss changes, weight distributions, and attention patterns of each flow type are monitored in real-time to ensure the stability and effectiveness of the training process. When training instability is detected, the learning rate or weight balancing parameters are adjusted.

