# OpenReview forum: "UniPINNs: A Unified PINNs framework for Multi-task learning of diverse Navier-Stokes equations"
_ICLR.cc/2026/Conference — ICLR 2026 Conference Withdrawn Submission_

### Official Review · Reviewer_uryB · 2025-10-30

**Soundness:** 2
**Presentation:** 2
**Contribution:** 1
**Rating:** 2
**Confidence:** 5

**Summary:**

The current work proposed a physics-informed machine learning (PINNs) framework for solving Navier-Stokes equation problems by the idea of combining multi-task learning techniques, cross attention mechanisms, and adaptive weighting for terms in the loss function. To show the effectiveness of the proposed methodology, the authors performed machine learning experiments on the classical fluid dynamics problems of lid-driven cavity flow, pipe flow, and Couette flow in two dimensions. Error analysis and visual comparison are presented in the article.

**Strengths:**

I believe the strongest aspect of this work is the motivation and identifying a good problem (which is the limitation of PINNs) and an effort to solve it. The quality of the figures is good, and I appreciate it. The explanation of the machine learning framework is pleasant, especially the transformers (or attention mechanisms), which shows a good understanding of it by the author team (in contrast to some other papers, which do not even know how it works in practice).

**Weaknesses:**

The authors correctly stated the limitations of PINNs in the abstract:

“low data efficiency under sparse supervision, weak generalization ability across flow patterns, and high computational costs due to repeated training”

However, I believe that first, it has been a while since others tried to remove these limitations, and there have been some successes in that direction. For example, the following paper tried to resolve the needs of PINNs for retraining on any new geometry by combining PointNet and PINNs (published in 2022, three years ago):

https://doi.org/10.1016/j.jcp.2022.111510

And there have been so many other works, and the author did not address almost any of them.

More importantly, and beyond what I mentioned above, the effort of the authors to resolve these issues is unsuccessful (at least from what they presented). At the end of the day, the authors solved three extremely easy fluid dynamics problems in 2D, where the flow is laminar, steady, and the geometry of the computational domain is simply a rectangle.

From Figure 2 of the manuscript, it is easy to realize that the proposed network is much heavier than usual PINNs (with a few MLP layers). There is no comparison between the number of trainable parameters of traditional PINNs and the current work.

All in all, the manuscript strongly suffers from a lack of novelty and a lack of challenging test cases that could potentially convince the reader that something serious can be solved using this methodology.

**Questions:**

I proposed some questions in the previous box. Please see those.

---

### Official Review · Reviewer_hZut · 2025-10-31

**Soundness:** 3
**Presentation:** 3
**Contribution:** 2
**Rating:** 2
**Confidence:** 3

**Summary:**

This paper addresses the limitations of PINN in terms of low data efficiency, weak generalization, and high computational cost. A multi-task framework is proposed that 1) a cross-task attention module is applied for the knowledge interaction and transfer across different flow types and 2) the adaptive weight balancing strategy is used for training state of each flow type. Experiments have been conducted to show the effectiveness of the proposed method compared to extensive baseline models.

**Strengths:**

- The three limitations of single-task PINNs are critical. The idea of using multi-task learning in PINN is important.
- The methodology is well-structured.
- The experiments compared different PINNs on selected flow types.

**Weaknesses:**

- The methodology needs to be elaborated.
- The experiments were not well-justified. More ablation studies are needed.

**Questions:**

- Section 3.2: How does the framework guarantee that the shared backbone network learns the common features? Does the task-specific dedicated layer have different architecture designs or just different network parameters?
- From Section 3.4, it looks like the model is trained end-to-end, but Appendix A3 indicates that some of the modules in the framework are pretrained separately. Could the authors clarify this? For the dynamic weight calculation, is $t$ the time step of the flow dynamic or the iteration step?
- Section 4.3: Figure 4 shows that the optimization for different flows varies, which is contradictory to the statement of “balanced optimization”. Could the authors explain what would be possible reasons?
- Section 4.2: The reference to baseline models should be provided. The comparison of the proposed framework with baseline models is not fair. Task-specific models for the lid-driven cavity and Couette flow should also be compared. Also, for ALPINN, MMPDE-Net, and Single-Task PINN, are they trained on three flow types separately? If so, the authors are suggested to add a setting in which these models are pretrained on two flow types and fine-tuned on the other, as the proposed framework has the shared backbone with common features, whereas the baseline models are not.
- Section 4.5: The ablation study is not sufficient. Experiments should be added to justify whether the common features are learned in the shared backbone, and task-specific information is learned in each task-specific module.
- The paper claims that the framework can simultaneously handle multiple flow types. However, the framework requires the number of types known at both training and test time, making it limited in generalization to novel flow types. It would be good if the authors could discuss the universality of the proposed framework for any flow types.

---

### Official Review · Reviewer_mTmD · 2025-10-31

**Soundness:** 3
**Presentation:** 2
**Contribution:** 2
**Rating:** 4
**Confidence:** 4

**Summary:**

The paper introduces UniPINNs, a unified multi-task PINN framework that uses a shared backbone, combined with a cross-flow attention mechanism, to solve multiple Navier-Stokes problems simultaneously via a unified multi-flow learning framework. The model enjoys higher generalizability than other PINN architectures.

**Strengths:**

-	The problem tackled is well motivated, as PINN training is costly, and being able to train simultaneously across different flow types is an important achievement.
-	The introduction of a novel architecture based on a combination of cross-flow attention and adaptive weight balancing, solidified by ablation studies which justify the importance of each component.
-	Experiments showing the UNIPINN model and training framework enjoy the best performance among multiple PINNs.

**Weaknesses:**

-	Some areas lack clarity, such as the architecture of the single-task PINN that is tested in the ablation study, or more details about the various parts of the training procedures described in the appendix (number of epochs, learning rate…) for reproducibility.
-	No reflection about how the method would scale further than 3 flow families.
-	No analysis about the trade-off between training this polyvalent model vs training 3 specialized models.
-	Some important physics-informed models such as PINO were omitted from the experimental comparison, despite them having been studied on classic problems such as the Lid-driven cavity.
-	Typo in Figure 1. PDF instead of PDE.

**Questions:**

-	Could the authors describe more in depth the various components of the training procedures (learning rate, number of epochs, possible overfitting when fine tuning ?).
-	Does Fig. 4 correspond to the results of the joint-training phase ? It would be interesting to have some plots of the fine-tuning phase, and to see if this part of the training is very sensitive to the seed (as different parts of the network are trained in succession at each epoch as the training set is looped upon, in my understanding).
-	Could this method generalize better than single task PINNs to some new, unseen flow setting ? It would be an interesting point to show this.
-	How is the single-task PINN constructed ? Does it have a single task specific network ? What if, in th ablation study, we had a PINN pre-trained on all equations (with 3 specific networks), but then fine-tuned on only one equation ? (that is, a network having the same size as the UNIPINN presented, but that could only reliably solve one of the three settings). Would it work better than the combined UNIPINN ? Would it be worth having  3 such large networks ? It would be interesting to see a study of the possible trade-offs here.

---

### Note · Authors · 2025-12-13

I have read and agree with the venue's withdrawal policy on behalf of myself and my co-authors.